# Increased Rice Susceptibility to Rice Blast Is Related to Post-Flowering Nitrogen Assimilation Efficiency

**DOI:** 10.3390/jof8111217

**Published:** 2022-11-17

**Authors:** Mathias Frontini, Jean-Benoit Morel, Antoine Gravot, Tanguy Lafarge, Elsa Ballini

**Affiliations:** 1PHIM, INRAE, CIRAD, Institut Agro, University Montpellier, 34060 Montpellier, France; 2IGEPP, INRAE, Institut Agro, University Rennes, 35000 Rennes, France; 3AGAP, INRAE, CIRAD, Institut Agro, University Montpellier, 34090 Montpellier, France; 4PHIM, INRAE, CIRAD, Institut Agro, 34060 Montpellier, France

**Keywords:** *Magnaporthe oryzae*, nitrogen induced susceptibility, NUE, nitrogen uptake, rice

## Abstract

Reducing nitrogen leaching and nitrous oxide emissions with the goal of more sustainability in agriculture implies better identification and characterization of the different patterns in nitrogen use efficiency by crops. However, a change in the ability of varieties to use nitrogen resources could also change the access to nutrient resources for a foliar pathogen such as rice blast and lead to an increase in the susceptibility of these varieties. This study focuses on the pre- and post-floral biomass accumulation and nitrogen uptake and utilization of ten temperate japonica rice genotypes grown in controlled conditions, and the relationship of these traits with molecular markers and susceptibility to rice blast disease. After flowering, the ten varieties displayed diversity in nitrogen uptake and remobilization. Surprisingly, post-floral nitrogen uptake was correlated with higher susceptibility to rice blast, particularly in plants fertilized with nitrogen. This increase in susceptibility is associated with a particular metabolite profile in the upper leavers of these varieties.

## 1. Introduction

Rice is the staple food of many countries and the most-consumed unprocessed cereal in the world [1]. In order to achieve an optimal yield, most rice varieties have been selected during the green revolution under normal or high-nitrogen conditions to achieve maximum grain yield [2]. More recently, in order to limit environmental hazards due to massive nitrogen utilization [3,4], breeders have developed modern rice varieties optimizing nitrogen use efficiency (NUE). NUE is commonly determined as the amount of yield (or biomass) by available nitrogen [5,6]. Nitrogen use efficiency can be subdivided into two components: (1) N-uptake efficiency (ability to uptake the N) and (2) N-utilization efficiency (the ability to convert the N uptake into grain dry matter yield) [5,6]. The N-uptake efficiency is usually broken down according to the date of flowering [7]. Another additional trait related to NUE is the Nitrogen remobilization efficiency which is the major component of N-utilization efficiency [8]. Genotypic differences in NUE have been largely documented in rice [2,9,10], but the genotypic diversity of NUE components has been poorly documented, [11] particularly Nitrogen uptake efficiency after flowering [12]. 

Despite its benefits to crop productivity described above, nitrogen fertilization is also known to alter plant susceptibility in many pathosystems [13,14,15] and in particular against the rice blast fungus *Magnaporthe oryzae* [16,17,18]. Beyond its impacts on the plant canopy which enhance epidemics [19,20], nitrogen also causes physiological changes within the plant which results in changes in plant/pathogen interactions [18]. An early hypothesis explaining this nitrogen-induced susceptibility (NIS) in rice was that the nitrogen supply caused cell wall thinning, which facilitated the penetration of *M. oryzae* appressorium [21]. However, although the impact of nitrogen on cell walls has been confirmed recently [9], the involvement of this process in *M. oryzae* penetration seems unlikely [16]. On the other hand, this study highlighted an increased aggressivity of *M. oryzae* and a modification of the level of expression of *OsGS1-2*, a gene encoding a Glutamine synthetase (GS) isoenzyme [18]. The fact that this enzyme is also one of the key enzymes involved in nitrogen use efficiency [22] led us to hypothesize that a link between NUE and higher susceptibility to rice blast may exist, particularly after fertilization.

Several genes involved in the regulation of NUE have been identified as having a role in the alteration of disease resistance, especially after nitrogen fertilization. For example, the balance between glutamine and glutamate is not only essential for nitrogen remobilization [23], but the impact of glutamate has also been shown in several pathosystems [24]. For example, a mutation in the *Fd-GOGAT* gene resulted in increased resistance to bacterial blight [25]. Ammonium uptake and assimilation are also essential for rice sheath blight resistance [26]. Amino acid homeostasis seems to be a major regulator of plant immune response [27,28].

This leads us to hypothesize that cultivars with different nitrogen use profiles may respond differently to nitrogen fertilization, resulting in different metabolic profiles and thus potentially to a variable susceptibility to rice blast. Our objectives were first to identify different nitrogen use profiles within a set of japonica rice. For this purpose, we have tested whether the sugar and amino acid compositions of individual leaves vary between these cultivars, and we have analyzed how these profiles change with the addition of nitrogen fertilization. These metabolite measurements and their variations have allowed us to estimate the physiological status and profiles depending on the NUE component ability of the selected varieties. We were then able to test our hypothesis to see if there was a correlation between the characteristics of these varieties and their higher susceptibility to rice blast after nitrogen fertilization. Finally, we were able to link nitrogen-induced susceptibility to post-flowering uptake efficiencies, one of the components of NUE.

## 2. Materials and Methods

### 2.1. Greenhouse Experimentation 

This experiment was conducted two times at the CIRAD Lavalette campus in Montpellier from April to August 2018 and 2019. Ten temperate japonica rice varieties were selected (Appendix A). The experiment was composed of three experimental groups (Appendix A). Each group was dedicated to a different kind of analysis and was composed of the ten varieties with eight or twelve replicated pots. The purpose was to have similar growth conditions for each group so as to be able to compare the results of each analysis. Plants were grown in 3.5 L pots with a homemade composition potting soil adapted to rice growth in greenhouse (30% fine Baltic blond peat, 15% fine Baltic black peat, 10% pearlite, 31% coco peat, 5% clay, 9% volcanic sand, pH 4.5). This plotting is a nutrient-neutral medium, allowing for complete control of fertilization. Fertilization was set up the day before sowing by burying Basacote^®^ 6M (14-3-19 NPK, with 6.5 NO_3_ and 7.5 NH_4_) using 10.5 g of fertilizer in the pot, i.e., 1.7 g of nitrogen.

The climatic conditions were programmed to four time slots: 7 a.m.–12 a.m., 12 p.m.–6 p.m., 6 p.m.–10 p.m., 10 p.m.–7 a.m. The temperature was set at 28–29 °C during the 7 a.m.–12 p.m. and 6 p.m.–10 p.m. time slots. From 10 p.m. onwards, the temperature gradually decreased to 22 °C–23 °C and then gradually increased again from 7 a.m. The temperature was set at 29–30 °C for the 12 p.m.–6 p.m. time slot. The humidity was set at 65% during the day and 70% at night, from 10 p.m.–7 a.m. In addition, LED lighting was provided in support of natural light with a gradual switch on and off in the periods of 7 a.m.–12 p.m. and 6–10 p.m.

The seedlings were planted on 13 April 2018 and 8 April 2019. Four seeds of each genotype were sown in each pot per modality. After two weeks, the two least-robust seedlings from each pot were removed. The goal was to obtain two vigorous rice seedlings in each pot. After sowing, the pots were rearranged according to the split-plot experimental design [29]. Each year represented a supra-block which was further subdivided into three blocks based on experimentations (Appendix A). 

Daily overhead irrigation was conducted until the beginning of heading (50 days after sowing). From that point on, irrigation was carried out by capillary action, by filling the shelves in which the pots were stored so that the water level reached about 7 cm. The base of the pots was thus constantly partly immersed, and the surface of the pots constantly in contact with the air. Two weeks before grain maturity (80 days after sowing), the tables were dried, and irrigation was stopped to promote grain maturation.

At the tillering stage, the first group was separated into two batches. Differential fertilization was conducted on each batch of plants in the morning after four weeks of growth post-sowing. Two fertilizing solutions were applied with 200 mL per pot: the N treatment and the N+1 treatment. In order to promote the absorption of this fertilizer, no irrigation was conducted the day before. The N and N+1 solutions were made following previously published protocols [17]. The two solutions share the same composition of potassium, phosphorus, and micronutrients and differ only in their nitrogen composition. Thus, the N solution does not contain any nitrogenous element, while the N+1 solution contains 80 mg/L of amino-nitrate. The objective was to mimic an important fertilization within the context of a split fertilization as it can be practiced in the field. As this type of practice can lead to an increased susceptibility to diseases [30], it was necessary to understand its origin. This test in the laboratory allows for reproduction of the impact of the nitrogen supply on the physiology of the plant. Our objective was therefore to observe the impact of excessive nitrogen fertilization and not of a low-input crop. At the end of the experiment, the N modality corresponded to plants cultivated with 1.7 g of nitrogen/plant (3.5 g of fertilizer for 1 L of potting soil for which the daily diffusion of nitrogen is estimated at 14 mg) and the N+1 modality corresponded to plants cultivated under the same conditions as the N modality but with an additional 16 mg of nitrogen via the addition of a fertilizing solution. Both batches were inoculated 24 h after this fertilization. Group two was set up in the same way as block one. Tissues samples were collected from these plants 24 h after fertilization for molecular analyses. The two blocks were placed and randomized independently. Group three was used to collect samples for NUE analysis and was designed with a split-plot method. Here, an experimental bloc corresponds to a position in the greenhouse in one year. This group was conducted separately because (i.) the design was different and (ii.) plants were grown until the maturity stage whereas the two other groups were conducted until the tillering stage and the removal of plants at tillering could have caused shading changes in this group.

### 2.2. Inoculation and Symptoms Evaluation

The *M. oryzae* strain used for the inoculations was the French strain FR94. The inoculation procedure was conducted as described in [17]. Plants were inoculated the day after nitrogen treatment. Plants were transferred from the greenhouse to the inoculation chamber in the morning to be inoculated in the afternoon. Inoculation was conducted by spraying 50 mL of spore suspension on nine pots. Once the plants were inoculated, they were placed in a dark room at 100% humidity for 12 h, and then put in a phytotron for seven days until the symptoms are analyzed. Symptoms were measured seven days after inoculation. The affected leaves were scanned with a resolution of 600 dpi. The obtained scans were analyzed using LeafTool which was an R package developed in-house and is available on GitHub depository (https://github.com/sravel/LeAFtool; accessed on 11 July 2022). This software provides the number of lesions detected relative to the surface area for each leaf scanned. The actual NIS value is derived from the ratio between the number of lesions by area in N+1 and that obtained in N. 

### 2.3. Sampling for Molecular Analysis

For each genotype/nitrogen modality, we had at our disposal twelve plants distributed two by two in six pots. A biological replicate consisted of the sampling of three leaves of the same leaf stage from three different plants in different pots. Thus, for a nitrogen/genotype/stage of leaf modality, four biological replicates were performed. Only the central area of the leaves was taken. Once collected, the pool of three leaves was put into a tube and placed immediately in liquid nitrogen for metabolite extraction. Metabolomic measurements were performed on a restricted set of genotypes: SEPYA_005, SEPYA_007, SEPYA_010, and SEPYA_011. The samples were then sent to the “Plateau de Profilage Métabolique et Métabolomique” of INRAE in Rennes (France), which performed the determination of the main amino acids and sugars. Seventeen sugars and amino acids were significantly detected in the samples: alpha alanine, arginine, asparagine, aspartate, fructose, glucose, glutamate, glutamine, glycerate, glycine, myo-inositol, phenylalanine, proline, quinate, serine, threonine, and tyrosine. 

### 2.4. Sampling and Measures for NUE

Measurements were made at three key stages: tillering, flowering, and maturity. We established the tillering stage at four weeks after sowing. The flowering stage of a genotype was determined when 80% of the observed tillers showed emerged stamens on at least 2/3 of their panicles. Finally, the maturity stage was determined when 80% of the grains had become yellow and glassy. Since each sampling phase was destructive, a batch of 4 pots in 2018 were pre-identified for each phase. In our analyses, each individual corresponds to a jar in a given condition and block for a sampling. The measured variables correspond to either the sum or the average of the two plants in the same pot, as appropriate.

For each stage, plants in the same pot were cut at the base of the stems. For the flowering phases, the number of tillers present in the pot was counted, and then the plants in each sample were separated into green blades on one side, and sheaths, stems, and panicles (including those already visible) on the other. Leaf dry matter (LfDW) and stem dry matter (sheaths + internodes, StrDW) were weighed per pot after 72 h in a 60 °C oven for each sampling. Total aboveground dry matter was calculated as the sum of the dry matter of the blades and stems (sheaths + internodes + panicles if present). -s. Straw dry matter (StrawDW) was measured by the same method described above. The grains were separated from the stalk and then sorted according to their filling. We weighted the total of full grains (grain production, GP) and empty grains. 

Organ nitrogen levels (leaf %N_Lf, stem %N_St, straw %N_Str, and grain %N_Gr) were measured via near infrared spectrometry (NIRS). NIRS analysis was conducted on dried and ground samples on an NIRSystem5000 spectrometer (wavelengths 1100–2500 nm). All samples were scanned twice, and the spectra were averaged. Calibration equations were conducted using a CIRAD database built with 2000 samples of various plants enriched with the computing of NIRS measures and the chemical measure of N with the Kjedahl method on 34 of our samples. Each concentration of N used in this study was a prediction calculated from this calibration. Validation of the results showed a good precision of the predictions, with an R^2^ and SECV of, respectively, 1.1 and 0.93. The compilation of these rates with the biomasses from weighing allows for estimation of the gross amount of nitrogen contained in the plants. 

The amounts of total *N* uptake up to flowering ($N_{uptake}^{flo}$) and up to maturity ($N_{uptake}^{Tot}$) have been calculated by computing the NIRS results with the dry matter of plant organs.
$$N_{uptake}^{pre\;flo}=StDW\times \%N.St+LfDW\times \%N.Lf$$
$$N_{uptake}^{Tot}=StrDW\times \%N.Str+GP\times \%N.Gr$$

The amount of *N* uptake between flowering and maturity was established by the calculation below
$$N_{uptake}^{post\;flo}=N_{uptake}^{Tot}-N_{uptake}^{flo}$$

The nitrogen remobilized was estimated by the difference between the total amount of *N* measured at flowering and the amount of *N* measured in the straw at maturity. This calculation assumes that most of the $N_{uptake}^{post\;flo}$ is transferred directly into the grains, the rest being negligible.
$$NRemo=N_{uptake}^{pre\;flo}-StrDW\times \%N.Str$$

### 2.5. Statistical Analysis

The first group’s results were analyzed using the following models: $$Yn=\mu +\gamma k+\alpha i+\beta j+\left(\alpha \beta \right)ij+\varepsilon n$$

With being *Y_n_* the number of observed lesions per unit area; *μ* being the theoretical mean, *γ_k_* being the Year effect; *α_i_* being the genotype effect; *β*_j_ being the nitrogen treatment effect; *(αβ)_ij_* being the genotype: nitrogen; and finally, *ε_n_* being the residual effect. In that model the statistical replicate corresponds to a scanned leaf. In these groups, the statistic individual is the scanned leaf. Thus, we worked with 20 repetitions for each genotype*treatment combination (12 in 2018 and 8 in 2019).

The biochemical results analysis was conducted according to the following model:$$Yn=\mu +lk+\alpha i+\beta j+\left(\alpha l\right)ik+\left(l\beta \right)kj+\left(\alpha \beta \right)ij+\left(\alpha \beta l\right)ijk+\varepsilon n$$

With *Y_n_* being the concentration of the measured element; *μ* being the theoretical mean, *l_k_* being the leaf effect; *α_i_* being the genotype effect; *β_j_* being the nitrogen treatment effect; *(αl)_ik_* being the genotype:leaf effect; *(lβ)_kj_* being the leaf:nitrogen effect; *(αβ)_ij_* being the genotype:nitrogen effect; *(αβl)_ijk_* being the genotype:nitrogen:leaf effect; and finally, *ε_n_* being the residual effect.

In these groups, the statistical replicate is the pool of three leaves, so 4 repetitions for each genotype*treatment*leaf stage combination. We also performed a sparse partial least squares-discriminant analysis (sPLS-DA). This is a linear classification that allows for directed discrimination according to the pre-established groups in which each individual is declared [31].

The third group was analyzed following the model below: Yn=μ+ρk+αi+εn

With *Y_n_* being the measured or calculated physiological character; *μ* being the theoretical mean; ρ*_k_* being the bloc effect; *α_i_* being the genotype effect; and *ε_n_* being the residual effect. Here the statistical replicate corresponds to a pot, so 4 pots.

The analyses were conducted under R software. Linear models and fitted means from these models were compiled via the lme4 and emmeans R packages. The sPLS-DA were calculated using the mixOmics R package. Correlations and their significance have been calculated according to the Pearson method.

## 3. Results

The level of nitrogen-induced susceptibility varies within the rice panel, and is related to the diversity of post-flowering uptake efficiencies.

The rice blast symptoms analysis after differential fertilization showed a genotype x treatment interaction effect (0.00168 **). In particular, among the ten genotypes tested, two genotypes (SEPYA_005 and SEPYA_007) were significantly more susceptible in the N+1 condition, with an increase of 77% (and, respectively, 54%) in the number of lesions per surface (Figure 1). In contrast, two other genotypes (SEPYA_010 and SEPYA_011) showed an increased resistance in the N+1 condition. The other six genotypes were not significantly impacted by nitrogen fertilization, confirming our previous results that nitrogen-induced susceptibility is a relatively rare phenotype in rice temperate japonica [17]. 

In the same experiment, the rice panel showed a diversity in nitrogen uptake efficiency in particular in post-flowering uptake efficiency (Figure 2). Interestingly, we were able to identify a significant correlation between post floral uptake efficiency and the level of nitrogen-induced susceptibility (Figure 3). However, although this correlation is significant, the R^2^ level is moderate but correct for such biological data. Thus, varieties with late-cycle nitrogen uptake also show a greater increase in rice blast susceptibility after nitrogen fertilization at the seedling stage.

This observation suggested that these varieties may already display different strategies of nitrogen utilization at seedling stage that could reflect the values of certain NUE components. To verify this hypothesis, four genotypes (SEPYA_005, SEPYA_007, SEPYA_010, and SEPYA_011) were chosen according to their contrasting behaviors in order to measure the molecular and physiological changes induced by nitrogen fertilization in early developmental stages.

### A Metabolic Profile Is Associated with Nitrogen Induced Susceptibility

An sPLSDA was performed, calculated from all of the molecular data acquired on the different leaf levels (Appendix A). We could identify contrasts between genotypes, but the nitrogen treatment groups did not show a clear separation. However, both NIS phenotypes (SEPYA_005 and SEPYA_007) were clustered on one side of the projection opposite the “No NIS” phenotype (SEPYA_010 and SEPYA_011). This separation of NIS and No-NIS individuals appeared to be based in particular on myo-inositol levels, higher in SEPYA_005 and SEPYA_007, independently of leaf levels. Other variables helped to describe this separation but are dependent on leaf level. Myo-inositol levels are opposite to glutamine, alpha alanine, aspartate, and tyrosine levels. The second dimension of this projection was described by arginine and glycine opposing methionine and isoleucine. 

The addition of nitrogen induced variations in the concentration of metabolites correlated with a change in susceptibility: aspartate, glucose, quinate, and threonine (Figure 4). For example, the higher the glucose, fructose, and asparagine levels in the leaves, the less susceptible they were. Thus, it appeared that a plant showing an NIS phenotype tended to have leaves that were less rich in these metabolites after fertilization. However, these correlations did not inform us about the impact of nitrogen treatment on these variables.

## 4. Discussion

The first major result of our study is the relationship between the induction of susceptibility caused by a nitrogen application (NIS) to four-week-old seedlings and the ability of the genotypes to uptake nitrogen after flowering. This correlation is intriguing, because it links a single event that occurs very early in the cycle (NIS) to a process (NUE) that is the result of the sum of many factors. One of these factors is the amount (and therefore the strength) of sink set up by the plant. Indeed, genotypes with the highest post-floral removal capacities are those with the highest production and maintenance of tillers (sinks). The two NIS genotypes (SEPYA_005 and SEPYA_007) were described as cultivars with higher production capacities than others when nitrogen was available at the end of the cycle. Conversely, cultivars such as SEPYA_010 and SEPYA_011 favored their pre-flowering nitrogen uptake. These cultivars may favor storage for late remobilization, generating too little sink strength at the end of the cycle to continue nitrogen uptake. These cultivars, which are not very efficient at the end of the cycle, do not show the NIS phenotype overall, and SEPYA_010 even shows a phenotype where the addition of nitrogen causes an increase in resistance. Finally, the other genotypes show rather intermediate N use trends, without showing NIS, suggesting that only those whose nitrogen uptake is preferentially post-floral are impacted in their susceptibility. These observations suggest that genotypes react differently to the 4-week nitrogen shoot depending on their nitrogen-use pattern throughout their cycle. This particularity could either come from their physiological reaction to the addition of nitrogen, but also from their root capacity to uptake this additional nitrogen. Root volume often develops in order to adapt to stress and at the expense of aboveground biomass [32]. Our hypothesis is that the addition of nitrogen could be related to the use of nutrients for tillering production and maintenance. Regulation of tillering in rice by nitrogen addition has already been shown in Japonica via expression of the *OsNPF7.7-2* gene in meristems [33]. This regulation could be specific in SEPYA_005 and SEPYA_007 and causing NIS. 

The other striking result is the implication of myo-inositol as a marker of NIS. This molecule, derived from glucose, is involved in several lipid signaling mechanisms, including membrane formation, auxin perception, stress response, and regulation of cell death [34]. This molecule is also involved in the synthesis of ascorbate, which scavenges reactive oxygen species and protects cells and organelles from oxidative damage [35]. The implications of this molecule in the defense mechanisms against rice blast remain unknown and contradictory. Indeed, Madhavan et al. [36] showed that the production of myo-inositol was higher in cell cultures of a susceptible rice variety seven days after putting them in the presence of *M. oryzae* elicitors. In contrast, this molecule appeared to accumulate more in resistant transgenic rice [37]. However, what these studies have in common is a variation in myo-inositol of several days post infection without knowledge of the initial amounts in the tissues. It would be interesting to see how these amounts vary over time after fertilization followed by inoculation.

Finally, we observed that the additional nitrogen fertilization induces a modification in four metabolites: aspartate, glucose, quinate and threonine. It seems that a plant showing an NIS phenotype will tend to have leaves less rich in these metabolites. Like many other fungi, the virulence of *M. oryzae* is modulated by the availability of sugar in the environment [38,39,40]. From these studies it would appear that *M. oryzae* would have an ability to adapt to the depletion of resources in its environment [38]. In our study it appeared that NIS varieties (i) had lower levels of glutamine, the preferred amino-acid of *M. oryzae*, than the others and (ii) underwent a greater drop in their glucose levels associated with their increased susceptibility. This suggests that the fungus is progressing in NIS varieties under unfavorable conditions like previously reported [41], which is paradoxical given the observed phenotype.

However, NIS response is probably not due to a single factor, but to a combination of changes. Indeed, the *M. oryzae*-rice interaction involves many metabolic changes in the host due to the struggle for nutrients and the use of metabolites in defense mechanisms [42,43]. As a result, many mechanisms could conflict with the remobilization message that the additional nitrogen fertilization triggers. This message could then either promote the fungal control responses (SEPYA_010), or conflict with this control resulting in NIS (SEPYA_005 and SEPYA_007). This disruption of the signal could completely change the expression of the measured defense genes or modify the aggressiveness of *M. oryzae*.

## 5. Conclusions

We were able to show a correlation between increased susceptibility to blast after nitrogen fertilization in varieties with better nitrogen uptake efficiency after flowering. This increased susceptibility could be related to different metabolite compositions in these varieties. It would be interesting to analyze the post-flowering nitrogen uptake efficiency in a larger number of varieties and to verify by which mechanisms these varieties modify their leaf molecular compositions as early as the seedling stage in order to understand why these varieties create a favorable environment for foliar pathogens such as *M. oryzae*.

## Figures and Tables

**Figure 1 jof-08-01217-f001:**
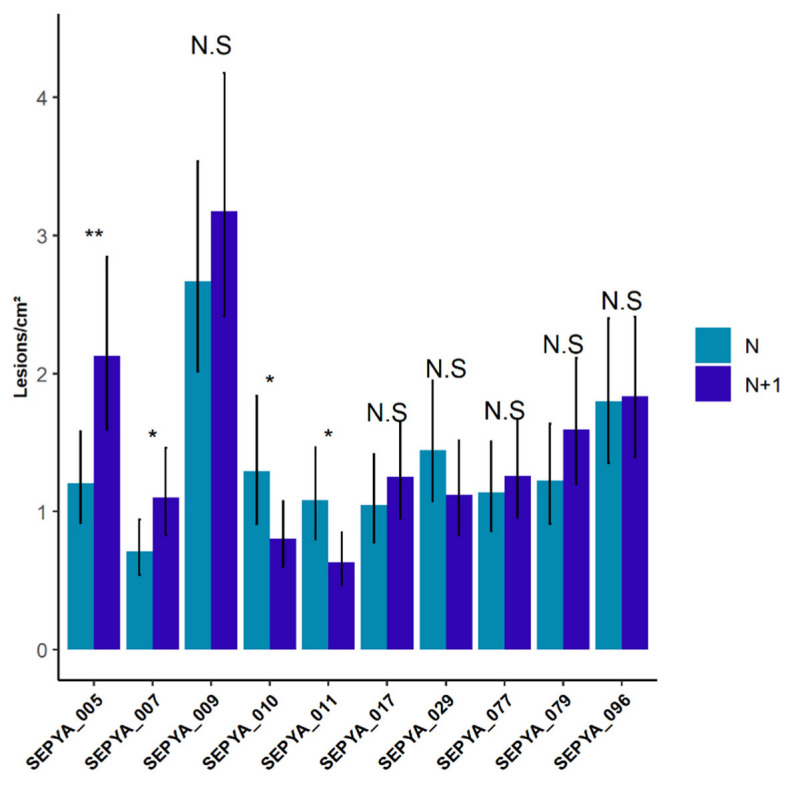
Variation in rice blast susceptibility depending on the nitrogen fertilization (N or N+1) applied one day before inoculation. The susceptibility was evaluated seven days after inoculation by counting the number of lesions per cm^2^. Values shown are the adjusted means and standard errors of the number of lesions per cm^2^ of the last ligated leaf in 2018 and 2019 calculated from the model described in Materials and Methods. Each contrast was calculated by a paired *t*-test of the genotypes and corrected by a Tukey adjustment. N: no nitrogen shoot, N+1: nitrogen shoot. Significance established as *p*-value > 0.1 NS; *p*-value < 0.1 Sub.S; *p*-value < 0.05 *; *p*-value < 0.01 **.

**Figure 2 jof-08-01217-f002:**
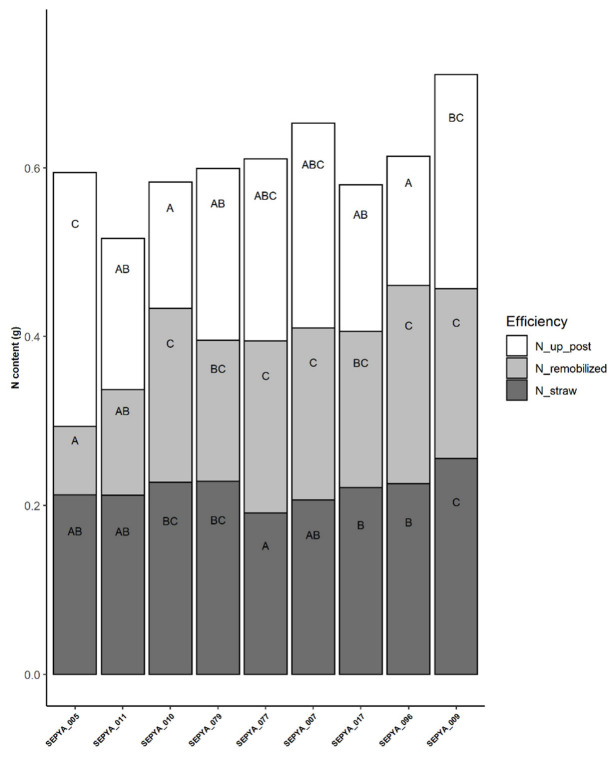
Diversity of nitrogen uptake and utilization in a panel of nine genotypes. Nitrogen uptake and partitioning between remobilized and fixed N share in straws. The sum of the remobilized nitrogen share and the fixed nitrogen share in mature straws is the pre-flower nitrogen uptake. The groups are derived from post-hoc tests performed on the model described in Materials and Methods, and are calculated by modality. N_straw: amount of nitrogen remaining in the straw at harvest, N_remobilized: amount of nitrogen remobilized, N_up_post: amount of nitrogen removed after flowering. The repetition of the data was not sufficient for SEPYA_029, which is not represented here. Letter categories have been established with post hoc test done with the linear model described in methods with a Tukey adjustment. Two bars of the same color with the same letter indicate no significant (*p* < 0.05) difference between component values.

**Figure 3 jof-08-01217-f003:**
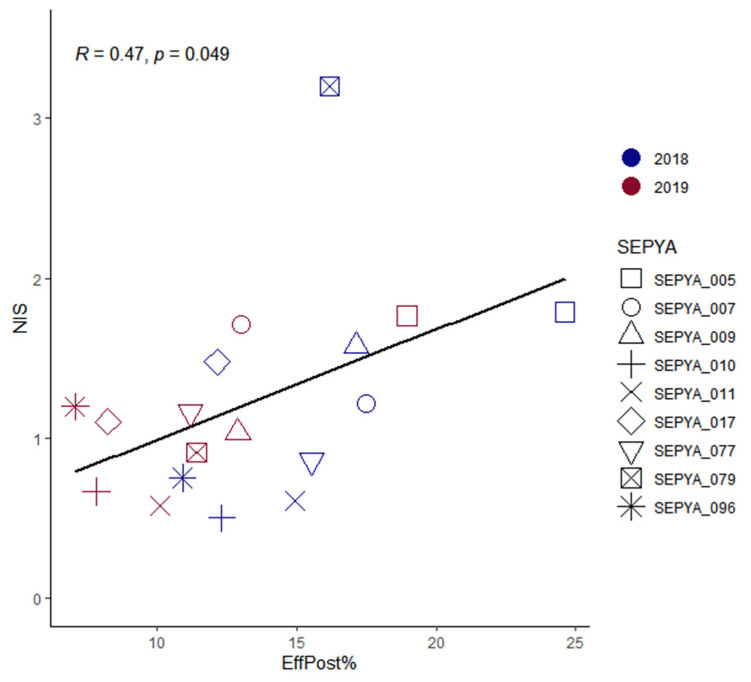
Relationship between NIS and the nitrogen-uptake efficiency after flowering. Each point represents the NIS value estimated from the susceptibility data as well as the amount of nitrogen removed after flowering. The points circled in blue are the 2018 estimates, and those circled in red are the 2019 estimates. The correlation value was determined using Pearson’s method.

**Figure 4 jof-08-01217-f004:**
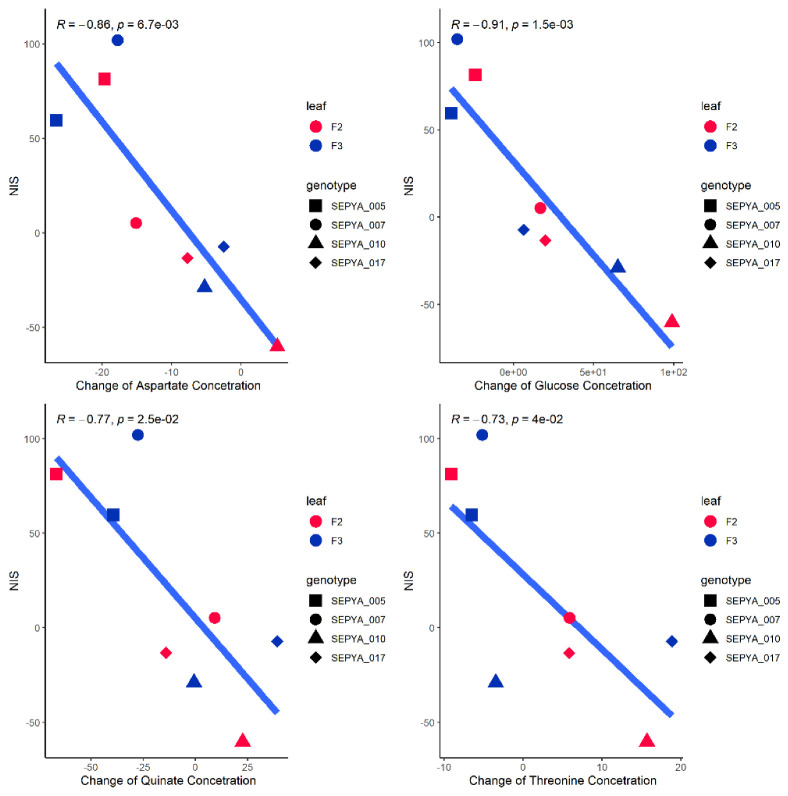
Relationships between nutrient drop and metabolite concentration changes following nitrogen supply and NIS. Correlation between changes in aspartate (top left), glucose (top right), quinate (bottom left), and threonine (bottom right) levels following nitrogen supply and NIS. Red dots are from measurements of F2 leaves, and blue dots are from F3 leaves. Correlations and significance were calculated using Pearson’s method.

## Data Availability

The data presented in this study are available on request from the corresponding author. The data are not publicly available due to protection of the data by the rice breeders involved in the project in order to protect the commercial interests.

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
