# Peer review of "Increased Rice Susceptibility to Rice Blast Is Related to Post-Flowering Nitrogen Assimilation Efficiency"

_jof, 2022, doi:10.3390/jof8111217_

Round 1

Reviewer 1 Report

No clear objectives. The manuscript needs substantial revision

Author Response

We are sorry if our manuscript does not meet the expectations. We have modified the introduction to hopefully make our objectives clearer and we have modified the materials and methods to clarify

Reviewer 2 Report

In the present manuscript entitled “Increased rice susceptibility to rice blast is related to post flowering nitrogen assimilation efficiency”, authors have investigated pre and post floral biomass accumulation and nitrogen uptake and utilization of ten temperate japonica rice genotypes and the relationship of these traits with molecular markers and susceptibility to rice blast disease. The need of the research is not well justified, methods have insufficiently explained, results are not understandably written and not discussed in the light of existing knowledge in the subject. Based on my evaluation, I have the following major suggestions that may be addressed.

Please provide the justification, why do you use nutrient-neutral potting soil at the mentioned concentration?

Please elaborate all the used protocols in more detail in the materials and methods sections.

Please add the statistical analysis section methodology section. How many replicates were conducted in each experiment?

The author has used some numeric values with 1 digit after decimal and some values without decimals.  I would recommend keeping uniformity in using the expressions.

Line-282-313, Please avoid repeating the results in discussion section.

Conclusion should be more compact with key features of their research instead of generalization.

Please provide the limitations of the study, as well as the future directions of research.

References are not well presented; these have several mistakes.

Author Response

We are sorry if our manuscript does not meet the expectations. We have modified the introduction to hopefully make our objectives clearer and we have modified the materials and methods to clarify.

We've try yo answer to the different suggestions:

  • Please provide the justification, why do you use nutrient-neutral potting soil at the mentioned concentration?  We've modified methods to better explain this part.
  • Please elaborate all the used protocols in more detail in the materials and methods sections. We've modified methods to better explain this part.
  • Please add the statistical analysis section methodology section. How many replicates were conducted in each experiment? We've modified methods to better explain this part.There are two year experiments with eight replicated pots each year. But for RNA and metabolites sampling a biological replicate consisted in the sampling of three leaves of the same leaf stage from three different plants in different pots. Thus, we have four biological replicates for molecular analysis for each year.
  • The author has used some numeric values with 1 digit after decimal and some values without decimals.  I would recommend keeping uniformity in using the expressions. I'm not sure I could identify which numbers were mentioned here
  • Line-282-313, Please avoid repeating the results in discussion section. We've tried to remove results from this paragraph
  • Conclusion should be more compact with key features of their research instead of generalization. Please provide the limitations of the study, as well as the future directions of research. The conclusion seems already compact to us and the suggestions that were proposed are the future directions of our research
  • References are not well presented; these have several mistakes. We've tried to find the mistakes but they were not obvious to us

Reviewer 3 Report

The authors discovered an exciting correlation between nitrogen-induced susceptibility to post-flowering uptake efficiencies. They found two opposite genotypes, which showed susceptibility and resistance to M. oryzae after nitrogen fertilization. The metabolic profile indicates that the increased susceptibility could be related to different metabolite compositions in these varieties. Overall, the manuscript is well-written and clear. However, I think the authors could plan and design other experiments to claim the discovery. I have the following comments on this work:

There exists intimate interaction between M. oryzae-rice during the infection. It is reported that the physiology and pathogenesis of fungi also respond to nitrogen application. So, the nitrogen-induced susceptibility of the host plant is directly or indirectly associated with nitrogen uptake efficiencies? Have you tested the virulence of M. oryzae in response to different nitrogen treatments? 

Line 105, add a reference.

Line 227, it is interesting to find two different genotypes that showed increased susceptibility and resistance in the N+1 condition. Have you taken pictures of the infection leaves? Could you provide the figures in your manuscript? 

Measuring the lesion size is a traditional way to evaluate the rice blast symptom. In addition to the number of lesions per surface, have you quantified the lesion size in different genotypes? 

I am curious about the nitrogen treatments: N and N+1. Why did you design these two treatments? Have you considered adding the nitrogen deficiency treatment (maybe named N-1) and comparing the results under N, N+1, and N-1 conditions?

In figure 4, did you apply statistical analysis to evaluate the significance of the correlation?

Line 55, Line 312, have you tested some gene expressions involved in the disease resistance and genes related to key enzymes for Nitrogen Use Efficiency?

Since mutual influence existed between different nutrient assimilation, have you tested other micro-or macro-nutrient concentrations in the plant leaves after different nitrogen treatments in response to pathogen infection? 

Author Response

We thank you for your suggestions. We have tried to answer them and to modify our manuscript to make our method clearer. Here are some of the modification we've proposed:

  • It is reported that the physiology and pathogenesis of fungi also respond to nitrogen application. So, the nitrogen-induced susceptibility of the host plant is directly or indirectly associated with nitrogen uptake efficiencies?Yes more precisely it's one of the component of NUE (the post flowerinf nitrogen uptake that is associated with NIS)
  • Have you tested the virulence of M. oryzae in response to different nitrogen treatments?  Yes we did it previously in another paper Huang et al. 2017.
  • Line 105, add a reference. Reference added
  • Line 227, it is interesting to find two different genotypes that showed increased susceptibility and resistance in the N+1 condition. Have you taken pictures of the infection leaves? Could you provide the figures in your manuscript?  Yes we had pictures but they were directly analysed using the software and because of the image size we were not able to keep all the pictures after analysis so we choose to not illustrate the picture.
  • Measuring the lesion size is a traditional way to evaluate the rice blast symptom. In addition to the number of lesions per surface, have you quantified the lesion size in different genotypes?  Yes the software is doing it but there were no effect on lesions size. Because the software can not necessarly distinguish coalescent lesions (two different lesions that are so closed that they merged) it's probably not suitable for this analysis and we considered this data not suitable.
  • I am curious about the nitrogen treatments: N and N+1. Why did you design these two treatments? This protocol is an adaptation of a protocol without basacote. At that time we were using a liquid fertilization each week (Ballini 2013). But for this experiment we used bigger pots and with the previous protocol plants were pale and could not be representative. That's why we decided to use Basacote to have enough fertilization until harvest. For the N+1 fertilization the idea was to mimick a tillering stage fertilization and to see if plants become more susceptible juste after this fertilization.
  • Have you considered adding the nitrogen deficiency treatment (maybe named N-1) and comparing the results under N, N+1, and N-1 conditions? We tried to use plants conducted in N/2 to test nitrogen shoot effect but we could not find any results. To do well it would be necessary to leave the fertilizer delay on the surface of the pot and to remove it before the shoot to have the modality N-1. But we were not able to try it.
  • In figure 4, did you apply statistical analysis to evaluate the significance of the correlation? Yes it's a pearson correlation that has been added to the method
  • Line 55, Line 312, have you tested some gene expressions involved in the disease resistance and genes related to key enzymes for Nitrogen Use Efficiency? Yes we tested several genes involved in defense, primary metabolism and NUE. Some gene like OsAlaAT are very significant but most of the gene had no significant results. We decided to not use these results in this paper.
  • Since mutual influence existed between different nutrient assimilation, have you tested other micro-or macro-nutrient concentrations in the plant leaves after different nitrogen treatments in response to pathogen infection?  Yes 17 metabolites are followed (see method) but in the results we show only the significant metabolites.